

# Curcumin effect on *Acanthamoeba triangularis* encystation under nutrient starvation

Rachasak Boonhok[1], Suthinee Sangkanu[2], Suganya Phumjan[1],
Ramita Jongboonjua[1], Nawarat Sangnopparat[1],
Pattamaporn Kwankaew[1], Aman Tedasen[1], Chooi Ling Lim[3],
Maria de Lourdes Pereira[4], Mohammed Rahmatullah[5],
Polrat Wilairatana[6], Christophe Wiart[7], Karma G. Dolma[8],
Alok K. Paul[9], Madhu Gupta[10] and Veeranoot Nissapatorn[2]

[1] Department of Medical Technology, School of Allied Health Sciences, and Research Excellence Center for Innovation and Health Products (RECIHP), Walailak University, Thai Buri, Nakhon Si Thammarat, Thailand
[2] School of Allied Health Sciences, Southeast Asia Water Team (SEA Water Team) and World Union for Herbal Drug Discovery (WUHeDD), and Research Excellence Center for Innovation and Health Products (RECIHP), Walailak University, Thai Buri, Nakhon Si Thammarat, Thailand
[3] Division of Applied Biomedical Science and Biotechnology, School of Health Sciences, International Medical University, Kuala Lumpur, Malaysia
[4] CICECO-Aveiro Institute of Materials and Department of Medical Sciences, University of Aveiro, Aveiro, Portugal
[5] Department of Biotechnology and Genetic Engineering, University of Development Alternative, Dhaka, Bangladesh
[6] Department of Clinical Tropical Medicine , Faculty of Tropical Medicine, Mahidol University, Rachathewee , Bangkok, Thailand
[7] The Institute for Tropical Biology and Conservation, Universiti Malaysia Sabah, Kota Kinabalu, Sabah, Malaysia
[8] Department of Microbiology, Sikkim Manipal Institute of Medical Sciences, Sikkim, India
[9] School of Pharmacy and Pharmacology, University of Tasmania, Hobart, Tasmania, Australia
[10] Department of Pharmaceutics, Delhi Pharmaceutical Sciences and Research University, New Delhi, India

Corresponding authors
Polrat Wilairatana,
polrat.wil@mahidol.ac.th
Veeranoot Nissapatorn,
veeranoot.ni@wu.ac.th

## ABSTRACT

**Background:** Curcumin is an active compound derived from turmeric, *Curcuma longa*, and is known for its benefits to human health. The amoebicidal activity of curcumin against *Acanthamoeba triangularis* was recently discovered. However, a physiological change of intracellular pathways related to *A. triangularis* encystation mechanism, including autophagy in the surviving amoeba after curcumin treatment, has never been reported. This study aims to investigate the effect of curcumin on the survival of *A. triangularis* under nutrient starvation and nutrient-rich condition, as well as to evaluate the *A. triangularis* encystation and a physiological change of *Acanthamoeba* autophagy at the mRNA level.

**Methods:** In this study, *A. triangularis* amoebas were treated with a sublethal dose of curcumin under nutrient starvation and nutrient-rich condition and the surviving amoebas was investigated. Cysts formation and vacuolization were examined by microscopy and transcriptional expression of autophagy-related genes and other encystation-related genes were evaluated by real-time PCR.

**Results:** *A. triangularis* cysts were formed under nutrient starvation. However, in the presence of the autophagy inhibitor, 3-methyladenine (3-MA), the percentage of cysts was significantly reduced. Interestingly, in the presence of curcumin, most of the parasites remained in the trophozoite stage in both the starvation and nutrient-rich condition. In vacuolization analysis, the percentage of amoebas with enlarged vacuole was increased upon starvation. However, the percentage was significantly declined in the presence of curcumin and 3-MA. Molecular analysis of *A. triangularis* autophagy-related (ATG) genes showed that the mRNA expression of the ATG genes, ATG3, ATG8b, ATG12, ATG16, under the starvation with curcumin was at a basal level along the treatment. The results were similar to those of the curcumin-treated amoebas under a nutrient-rich condition, except *Ac*ATG16 which increased later. On the other hand, mRNA expression of encystation-related genes, cellulose synthase and serine proteinase, remained unchanged during the first 18 h, but significantly increased at 24 h post treatment.

**Conclusion:** Curcumin inhibits cyst formation in surviving trophozoites, which may result from its effect on mRNA expression of key *Acanthamoeba* ATG-related genes. However, further investigation into the mechanism of curcumin in *A. triangularis* trophozoites arrest and its association with autophagy or other encystation-related pathways is needed to support the future use of curcumin.

Curcumin, Encystation, Nutrient Starvation, Real-time PCR

# INTRODUCTION

*Acanthamoeba* spp. are free-living amoebas present in the environment, particularly in soil and water (*Siddiqui & Khan, 2012*). Several species of *Acanthamoeba* have been characterized (*Chelkha et al., 2020*), and most of the human pathogenic species are classified into T4 genotype, for example, *Acanthamoeba castellanii*, *A. polyphaga*, and *A. triangularis* (*Guimaraes et al., 2016*; *Juarez et al., 2018*; *Hussain et al., 2020*). *Acanthamoeba* spp. are transmitted to humans by different routes (*Neelam & Niederkorn, 2017*; *de Lacerda & Lira, 2021*) and lead to various clinical presentations, especially in immunocompromised individuals who may have granulomatous amebic encephalitis (*Matson et al., 1988*), chronic sinusitis (*Kim et al., 2000*), or cutaneous lesions (*Morrison et al., 2016*). In addition, for healthy individuals who wear contact lenses, this group is at risk of *Acanthamoeba* infection if they have poor personal hygiene habits and *Acanthamoeba* keratitis (AK), a well-known ocular disease caused by this protozoan parasite, usually present in the group of people (*Lorenzo-Morales, Khan & Walochnik, 2015*; *Neelam & Niederkorn, 2017*; *Khan, Anwar & Siddiqui, 2019*). Regarding *Acanthamoeba* life cycle, the amoeba usually presents a trophozoite stage, a metabolically active form and can multiply within the human host. However, after being exposed to a stressful condition, it can transform into a cyst form with a double wall that is more resistant to a harsh environment (*Anwar, Khan & Siddiqui, 2018*). This form is a major barrier to *Acanthamoeba* treatment. So far, several drugs have been approved by the

United States Food and Drug Administration, but standard therapeutic management of *Acanthamoeba*-infected patients is not yet available (*Elsheikha, Siddiqui & Khan, 2020*). The two most commonly used first-line drugs for *Acanthamoeba* treatment, especially in AK patients, still use chlorhexidine and polyhexamethylene biguanide. However, identifying new compounds and screening natural extracts for amoebicidal activity are still attractive approaches for further studies. It could provide an alternative drug for *Acanthamoeba* treatment or be used as a complementary treatment for *Acanthamoeba* infection in the future.

Autophagy is a lysosomal degradation pathway for intracellular cytosolic materials (*Yorimitsu & Klionsky, 2005*; *Feng et al., 2014*). This mechanism is essential for all eukaryotic cells to supply energy and support cell survival. In humans, a defect of the autophagy mechanism is associated with several diseases, for example, neurodegenerative diseases (*Menzies, Fleming & Rubinsztein, 2015*), non-alcoholic fatty liver disease (*Khambu et al., 2018*), or infectious diseases (*Castillo et al., 2012*; *Brinck Andersen et al., 2020*). Starvation or nutrient depletion is a classical stress condition for autophagy induction both *in vitro* and *in vivo* (*Mizushima et al., 2004*; *Suzuki, 2013*). Several autophagy-related (Atg) proteins participate in the formation of a double-membrane vacuole called autophagosome (*Eskelinen, 2005*; *Feng et al., 2014*). In mammals, more than 30 Atg proteins have been identified (*Feng et al., 2014*). However, a partial list of ATG genes is conserved in free-living amoeba, including *Acanthamoeba* spp., and some Atg proteins have been identified (*Picazarri, Nakada-Tsukui & Nozaki, 2008*; *Moon et al., 2009*; *Song et al., 2012*; *Kim et al., 2015*). *Acanthamoeba* autophagy is of interest as a number of Atg proteins have been partially characterized and reported to be involved with *Acanthamoeba* encystation, a mechanism in which trophozoites transform to cysts (*Moon et al., 2011*; *Song et al., 2012*; *Moon et al., 2013*; *Kim et al., 2015*). Hence, a study of autophagy at both transcriptional and protein levels is needed to understand its biological functions and interaction with other intracellular pathways, which further extends to its association with their pathogenesis in humans.

Curcumin, an active compound obtained from turmeric, *Curcuma longa* (*Kocaadam & Şanlier, 2017*), contains several pharmacological activities, for example, anti-inflammatory (*Wal et al., 2019*), antioxidant (*Jakubczyk et al., 2020*), anti-cancer (*Vallianou et al., 2015*; *Tomeh, Hadianamrei & Zhao, 2019*), and antimicrobial activities (*Cui, Miao & Cui, 2007*; *Martins et al., 2009*; *Teow et al., 2016*; *Yang et al., 2016*; *Mitsuwan et al., 2020*). The amoebicidal activity of curcumin against *A. triangularis* trophozoites and cysts was recently identified (*Mitsuwan et al., 2020*). It reveals another property of curcumin against this water-borne parasitic pathogen and could be a promising compound for further drug development against *Acanthamoeba* infection. In this study, we investigated the effect of curcumin on surviving *A. triangularis* trophozoites after being expose to a sublethal dose of curcumin under nutrient starvation and nutrient-rich condition. Cyst formation and vacuolization were examined by microscopic observation and molecular analysis of autophagy-related as well as other encystation-related genes at the transcriptional level, was investigated by real-time PCR. This raises another point of concern, in addition to the killing activity by plant extract or compound where surviving amoebas after the treatment
are likely to transform into a cyst, with an emphasis on *Acanthamoeba* autophagy, which is one of the pathways involved with *Acanthamoeba* encystation.

## MATERIALS AND METHODS

### *A. triangularis* cultivation

PYG medium, a nutrient-rich condition or full medium, (2% (w/v) proteose peptone, 0.1% (w/v) yeast extract, 400 $\mu$M $CaCl_2$, 4 mM $MgSO_4$, 2.5 mM $Na_2HPO_4$, 2.5 mM $KH_2PO_4$, 50 $\mu$M $(NH_4)_2Fe(SO_4)_2$, 100 mM glucose) was used to grow *A. triangularis* trophozoites, strain WU19001 (*Mitsuwan et al., 2020*). The parasite was maintained at room temperature (RT) without shaking (*Taravaud, Loiseau & Pomel, 2017*). The culture medium was replaced with fresh PYG every 2 days until trophozoites harvesting. To induce *A. triangularis* cysts, trophozoites were washed and grown in PAS supplemented with 5% glucose, a nutrient-depleted condition, called starvation, which was modified from Aqeel and colleagues based on our initial in-house laboratory trials (*Aqeel et al., 2013*). The PAS powder, obtained from HiMedia, Mumbai, India, consisted of NaCl, $MgSO_4 \cdot 7H_2O$, $CaCl_2 \cdot 2H_2O$, $Na_2HPO_4$ and $KH_2PO_4$.

### Curcumin preparation and determination of the half-maximal inhibitory concentration ($IC_{50}$)

Curcumin powder was commercially purchased (Sigma Aldrich, St. Louis, MI, USA). The curcumin was dissolved in 100% DMSO and prepared at stock 750 mg/mL. This was further diluted to the working concentration with medium. The identification of the $IC_{50}$ against *A. triangularis* trophozoites was performed in 96-well black plate (SPL Life Sciences, Seoul, Korea). Curcumin concentration was prepared with 2-fold serial dilution with starting final concentration of 8,000 $\mu$g/mL. Thus, the maximum of the final %DMSO was 2.13. Trophozoites were harvested and washed with fresh AnaeroGRO™ Peptone Yeast Extract Glucose Broth: Proteose peptone and yeast extract were purchased from HiMedia Laboratories, Mumbai, India. Sodium citrate dihydrate ($C_6H_5Na_3O_7 \cdot 2H_2O$), disodium phosphate ($NaHPO_4$), sodium chloride (NaCl), calcium chloride ($CaCl_2$), and glucose were purchased from Sigma Chemical Co. (St. Louis, MO, USA). Potassium dihydrogen phosphate ($KH_2PO_4$) and magnesium sulfate heptahydrate ($MgSO_4 \cdot 7H_2O$) were procured from Labscan (Bangkok, Thailand). Trypan blue (0.4%) was obtained from Gibco BRL (Grand Island, NY, USA). All chemicals and medium components used were of analytical grade and added at $2 \times 10^4$ cells/well. A control group of untreated cells and PYG medium alone was included. All edge wells were filled with Page's saline (PAS) buffer. After 24 h post-treatment, the parasite viability was analyzed by PrestoBlue® reagent (Invitrogen, Waltham, MA, USA) staining according to the manufacturer's protocol. The plate was incubated for 30 min at 37 °C incubators, and fluorescence intensity was measured at excitation/emission wavelength of 535/615 nm by a microplate reader (BioTek SynergyTMMX microplate reader, Winooski, VT, USA). The $IC_{50}$ of curcumin was then calculated by prism5 software (GraphPad Software, San Diego, CA, USA). The experiments were conducted in triplicate with three independent experiments.

## Analysis of cysts formation and vacuolization

*A. triangularis* trophozoites were cultured in PYG medium or PAS supplemented with 5% glucose, and the curcumin was added at a final concentration of 50 μg/mL for 24 h. After Trypan Blue staining, the parasites were assessed for cysts formation and vacuolization every 6 h after the curcumin treatment. At least 200 viable cells per condition were investigated, and different forms of the parasites *i.e.* irregular trophozoites, rounded trophozoites, and cysts, were identified under a light microscope. The percentage of cyst and proportion of parasite forms at each time point was calculated. The irregular trophozoites were further evaluated for their vacuole formation, and the surviving trophozoites of at least 100 cells per condition were analyzed. Trophozoites with vacuoles, regardless of their size, as well as the trophozoites containing enlarged vacuoles, were examined. An enlarged vacuole (EV) was defined as a vacuole with a diameter of at least 5 μm, and the trophozoite containing at least one EV was counted as 1 (*Boonhok et al., 2021b*). The experiment was performed with three independent experiments.

## Determination of minimal inhibitory concentration (MIC) and drug combination assay

A drug combination study of chlorhexidine, a standard anti-*Acanthamoeba* drug, and curcumin was performed for their amoebicidal activity. The minimum inhibitory concentration (MIC) of curcumin and chlorhexidine was identified along with the microtiter broth dilution method (*Mitsuwan et al., 2020*). The drug/compound was prepared in 96-well clear plate (SPL Life Sciences, Seoul, Korea), and trophozoites of $2 \times 10^5$ cells/100 μL were then added into each well. The plates were incubated at RT in the dark for 24 h. The parasite viability was quantified by Trypan Blue staining under a light microscope, Eclipse TE2000-S (Nikon, Konan, Minato-ku, Tokyo, Japan). The MIC value in our study referred to the lowest concentration with *A. triangularis* growth inhibition greater than 90%. Thus, the MIC of curcumin and chlorhexidine was 250 and 16 μg/mL, respectively. Their MICs were used in drug combination as-say as a starting concentration. The drug and curcumin were prepared in 96-well plate with 2-fold serial dilution. Then, the trophozoites were added and incubated at RT for 24 h before quantification of parasite viability. To examine the effect of curcumin in combination with autophagy inhibitors, 3-methyladenine (3MA) and wortmannin, which were purchased from Sigma Aldrich (St. Louis, MI, USA), the drug combination assay was then performed as mentioned above. However, the starting concentration of the inhibitors was used at 20 μg/mL to cover the concentration tested in this study.

## Preparation of total RNA and cDNA synthesis

*A. triangularis* trophozoites were cultured with PYG medium or PAS supplemented with 5% glucose medium in a 24-well transparent plate with a final number of $2 \times 10^5$ cells per well. Parasites were treated with curcumin at a final concentration of 50 μg/mL and incubated at room temperature for 24 h. Then, the parasites were harvested every 6 h after treatment (6, 12, 18, 24 h). Parasite preparation for RNA extraction and cDNA synthesis

were performed as described by Boonhok and his colleagues (*Boonhok et al., 2021a*). The cDNA was kept at −20 °C until use.

## Validation of PCR primers

All specific primers against *Acanthamoeba* genes used in this study are listed in Table S4. The target genes were ATG3 (GenBank accession no. GU270859), ATG8b (GenBank accession no. KC524507.1), ATG12 (GenBank accession no. HQ830265.1), ATG16 (GenBank accession no. FJ906697), cellulose synthase (CS) (GenBank accession no. EDCBI66TR), serine proteinase (SP) (GenBank accession no. EU365404), metacaspase (MCA) (GenBank accession no. AF480890), interleukin-1 converting enzyme-like protease (IL) (GenBank accession no. XM_004338552), and 18S rRNA was used as a reference gene. These primers were tested against *A. triangularis* strain WU19001 DNA. To confirm primer specificity, the PCR product was sent for sequencing (Apical Scientific Sdn. Bhd., Seri Kembangan, Selangor, Malaysia), and the DNA sequence was then analyzed and blasted against *A. castellanii* NCBI databases before performing a quantitative PCR (*Boonhok et al., 2021a*).

## Analysis of gene expression by quantitative PCR

iTaq Universal SYBR Green Supermix Kit obtained from Bio-Rad (Bio-Rad, Hercules, CA, USA) was used to prepare the quantitative PCR reaction (qPCR) along with the manufacturer's instruction. The 18S rDNA was used as a reference gene. The reaction was conducted in a PCR tube, which consisted of 10 μL of 2× iTaq Universal SYBR Green Supermix, 100 ng cDNA, and 1 μL of 200 mM F+R primers. The total volume was adjusted with DEPC water to 20 μL. Thermal cycler, StepOnePlus Real-time PCR systems (Applied Biosystems, Waltham, MA, USA), the software was set as follows; holding stage 95 °C for 30 s, cycling stage for 40 cycles at 95 °C for 15 s, 60 °C for 60 s, and melting curve stage at 95 °C for 15 s, 60 °C for 60 s, 95 °C for 15 s with a temperature increase of 0.3 °C. The average deltaCt ($\Delta$Ct) was obtained by the thermal cycler. The delta-delta Ct ($\Delta\Delta$Ct) and a relative expression of the mRNA were calculated as follows, the $\Delta\Delta$Ct = ((Ct of treated sample GOI-Ct of treated sample housekeeper) − (Ct of untreated control GOI-Ct of untreated control housekeeper)) where GOI refers to the gene of interest. The relative expression of mRNA = 2 to the power of (minus X) or $2^{-X}$ where X is $\Delta\Delta$Ct. The interpretation of the result is if the value >1, <1, and 1 means that the expression is increased, decreased, and constant, respectively.

## Statistical data analysis

Experiments were conducted with 2–3 technical replicates in three independent experiments. All data were recorded in Microsoft Excel 2016 (Microsoft Corporation, Redmond, Washington, USA). The statistical analysis was done using Prism five software (Graphpad Software, San Diego, CA, USA), and the mean ± SD or ± SEM was used, including a two-tailed unpaired Student's t-test. *P* values less than 0.05 were considered statistically significant.

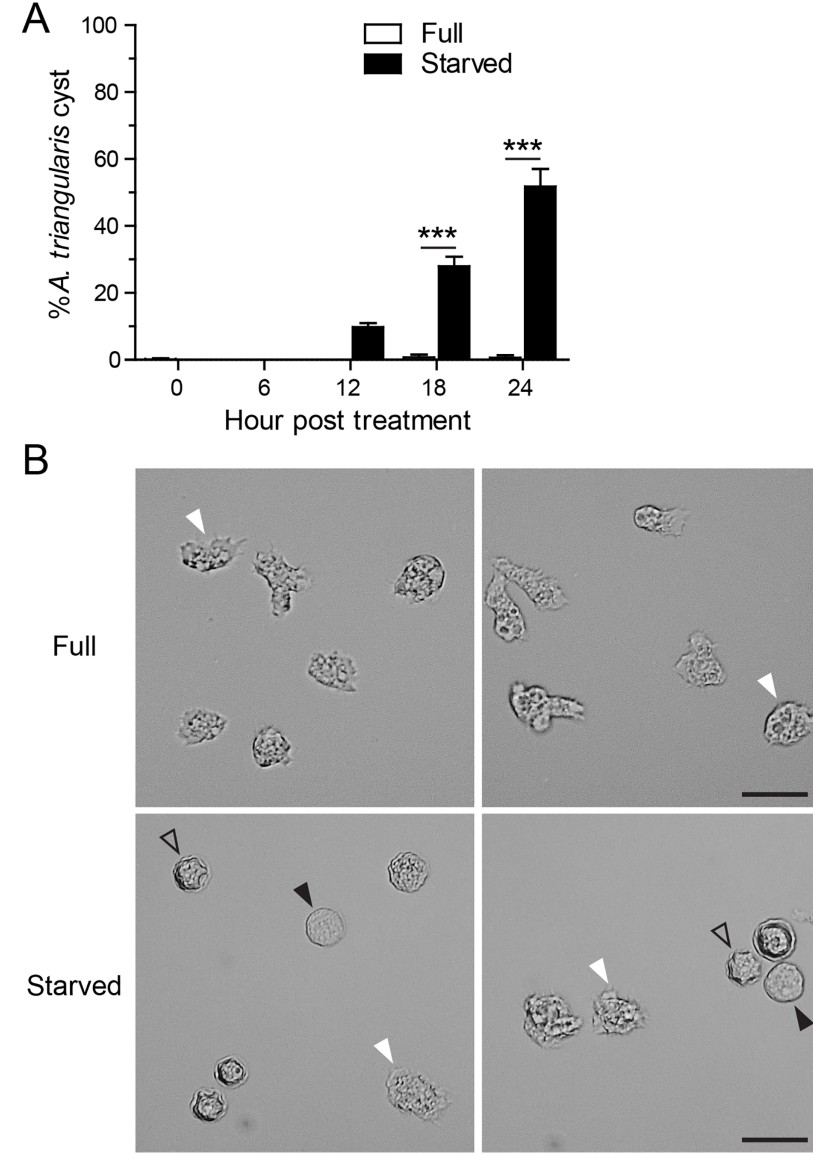

**Figure 1 *A. triangularis* cysts formation under starvation.** (A) The trophozoites were cultured in the starvation medium, PAS supplemented with 5% glucose, for 24 h. The parasites grew in PYG medium, or full medium was used as a control. The parasites were stained with Trypan Blue, and the viable parasites were analyzed under a microscope at the indicated time points. *A. triangularis* cysts were counted and represented as mean percentage ± SD. Data were obtained from three independent experiments. ***$P < 0.001$. (B) Representative images of parasites cultured in full and starved medium. Bar 20 μM. White and black arrowheads indicate the irregular trophozoites and rounded trophozoites, respectively, while the unfilled arrowhead indicates cysts.

## RESULTS

### Starvation induces *A. triangularis* encystation

Starvation or nutrient-depleted condition was used to induce *A. triangularis* encystation. The mean percentage of *A. triangularis* cysts was approximately 53.30%, and it was significantly different from that of the nutrient-rich condition or full medium (Fig. 1A).

The representative images of the parasites under starved and full conditions are shown in Fig. 1B.

## Effect of curcumin on *A. triangularis* autophagy under starvation

The $IC_{50}$ of curcumin against *A. triangularis* trophozoites under the full condition at 24 h treatment was first identified, and the $IC_{50}$ was 48.64 ± 30.86 µg/mL. The representative data is shown in Fig. S1. The curcumin concentration of 50 µg/mL was then used as representative curcumin concentration throughout this study. Starvation alone was included as a positive control for *A. triangularis* encystation. Starvation is known as a classical autophagy inducer in several eukaryotic cells (*Kamada, Sekito & Ohsumi, 2004*; *Díaz-Troya et al., 2008*); thus, in this assay, we included autophagy inhibitors, 3MA, and wortmannin, to examine a physiological and morphological change of *A. triangularis* upon the treatment. Our result showed that in the presence of 1 mM 3MA, the encystation was significantly decreased, whereas the 1 µM wortmannin was slightly impacted to the encystment. The percentage of cysts under starvation + 3MA or wortmannin was approximately 22.55% and 41.30%, respectively (Fig. 2A). We next investigated whether curcumin stress supports cysts formation under starved conditions. Interestingly, in the presence of 50 µg/mL curcumin, the percentage of cysts was approximately 1.39%, the surviving amoeba remained in the trophozoites stage (Fig. 2A). The representative images of the curcumin-treated parasite under starved conditions are shown in Fig. 2B. In addition, the parasites treated with a combination of curcumin and autophagy inhibitors were included in this experiment to test whether this could completely inhibit the encystation or not. The result showed that the percentage of cysts in curcumin + 3MA and curcumin + wortmannin remained at the basal level similar to that of curcumin-treated alone (Fig. 2A). Moreover, different viable forms of *A. triangularis*, namely, irregular trophozoites, rounded trophozoites, and cysts under different conditions, starvation alone, starvation + 3MA, and starvation + curcumin, were quantified under the microscope every 6 h. In starvation alone, the irregular trophozoites started to round off at 6 h. Cysts were seen 12 h after culture, and at 24 h, the percentage of cysts was approximately 50%. In the presence of 3MA, cysts were clearly seen at 18 h post-treatment, and at 24 h, the percentage of cysts was approximately 20%. Under curcumin-treated conditions, the parasites were mainly in irregular trophozoites, approximately 90%, and the percentage of cysts was approximately 2% (Fig. S2).

The evaluation of vacuolization in surviving trophozoites was further performed. The number of trophozoites containing vacuole and trophozoites with enlarged vacuole were analyzed under the light microscope. The percentage of trophozoites with vacuoles in the curcumin-treated condition was almost 100% along with the 24 h treatment, and the percentage was similar to that of starvation alone and starvation + 3MA (Fig. S3A). The percentage of trophozoites with enlarged vacuoles was further investigated. In starvation alone, the percentage increased significantly with the treatment, and the mean percentage at 24 h was approximately 22.41%. Interestingly, in the curcumin-treated condition, the mean percentage along the treatment was significantly reduced and

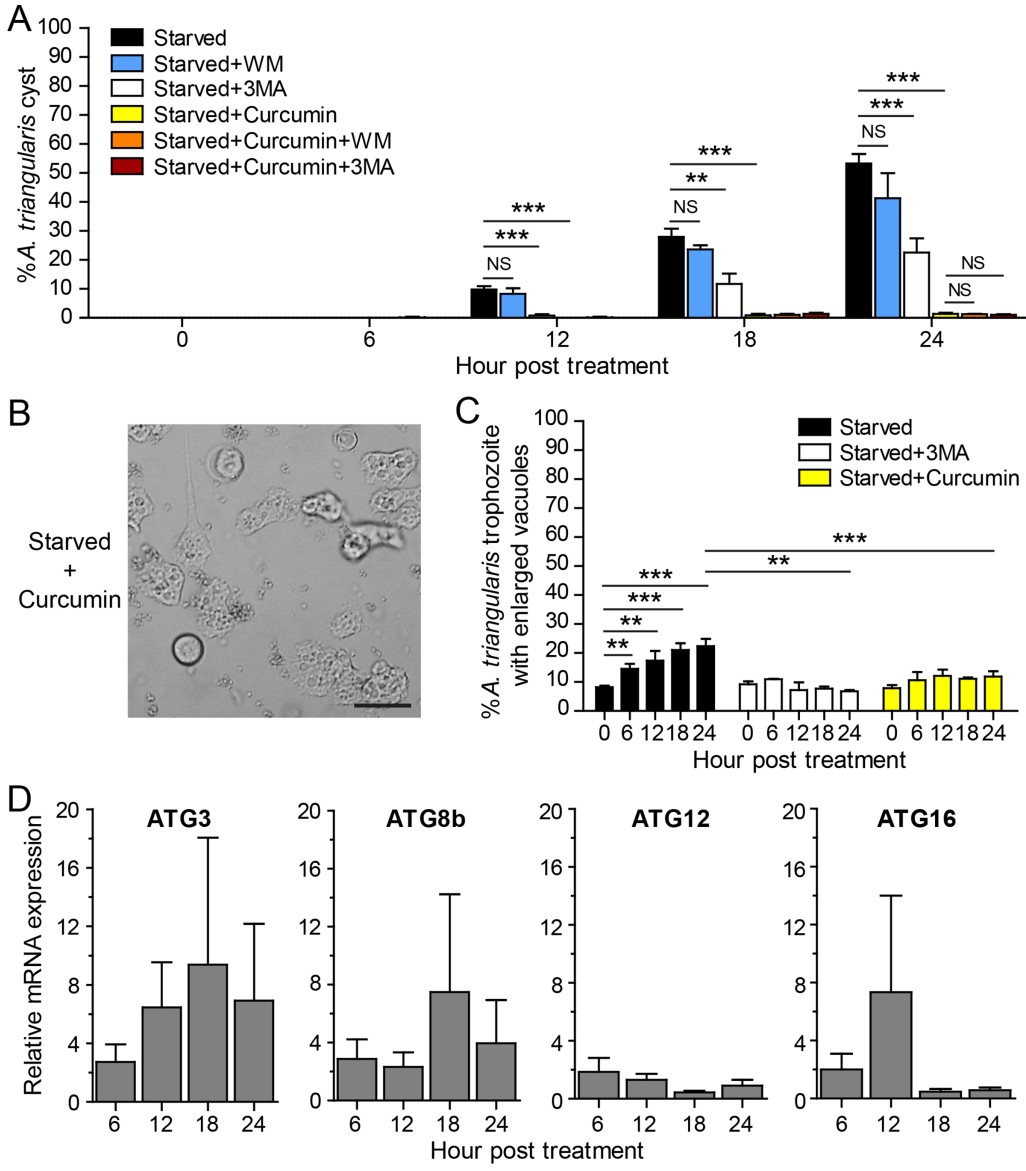

**Figure 2 *A. triangularis* response to curcumin under starved condition.** (A) Cysts formation, the trophozoites were cultured in a starvation medium, PAS+5% glucose, with autophagy inhibitors or 50 µg/mL curcumin ± autophagy inhibitors for 24 h. Starvation alone was included as a positive control. Cyst was quantified every 6 h post-treatment. The percentage of cyst was calculated and represented as mean ± SD. Data were obtained from three independent experiments. NS, not significant; ** $P < 0.01$; *** $P < 0.001$. (B) Representative image of curcumin-treated parasites under starved condition. Bar 20 µM. (C) Vacuolization in surviving trophozoites, at least 100 cells, the trophozoites per condition were examined for enlarged vacuole, a diameter of at least 5 µm. Data were obtained from three independent experiments and represented as a mean percentage ± SD. ** $P < 0.01$; *** $P < 0.001$. (D) Transcriptional expression of autophagy-related genes after curcumin treatment, *A. triangularis* trophozoites were cultured in starvation medium with or without 50 µg/mL curcumin for 24 h. The parasites were harvested every 6 h, and the mRNA level of ATG3, ATG8b, ATG12, ATG16 genes were analyzed by qPCR. Their expression at each time point was expressed as a relative mRNA expression. The 18S rRNA was included as a reference gene. The expression at time 0 h was set to 1. The data were obtained from three independent experiments. Bar graphs represent mean ± SEM.

maintained in the range of 7.89–12.16%. The result was similar to the 3MA-treated condition where the percentage was in the range of 6.79–10.92% (Fig. 2C).

The molecular analysis of *A. triangularis* autophagy-related genes, ATG3, ATG8b, ATG12, and ATG16, was conducted at transcriptional level upon curcumin treatment. The validation of PCR primers (Table S4) by conventional PCR against *A. triangularis* DNA was performed, and the target genes were successfully amplified. The gel result representing PCR products is shown in Fig. S4. In addition, analysis of DNA sequencing of the amplicons was performed, and the results are shown in Table S5. Then, the quantitative PCR was performed, and the results showed that the mRNA expression of all tested ATG genes was unchanged along with the treatment and maintained at the basal level (Fig. 2D). In addition, the expression of these ATG genes under 3MA-treated conditions was investigated. As expected, the expression of the ATG genes was at the basal level along with the treatment (Fig. S5). The overall results demonstrated the inhibitory effect of curcumin on the surviving amoebas against *A. triangularis* encystation even under starvation.

## Effect of curcumin on *A. triangularis* autophagy under a nutrient-rich condition

To measure the effect of curcumin alone without the stress of starvation, *A. triangularis* trophozoites were cultured in PYG, a nutrient-rich medium. As expected, curcumin did not activate cyst formation. The percentage of cysts was at a basal level and was not significantly different from that of the full medium alone (Fig. 3A). The representative image of the parasites under curcumin treatment is shown in Fig. 3B. The surviving parasites remained in the trophozoite stage.

Vacuolization in the surviving trophozoites was then analyzed. The percentage of trophozoites with vacuoles under curcumin treatment was nearly 100%, and the percentage was at a level comparable to full medium alone (Fig. 3C). To investigate the maturation of vacuole, parasites with enlarged vacuoles were examined. The percentage of trophozoites with enlarged vacuoles was consistent with the treatment, in the range of 6.23–6.81%.

Molecular analysis by qPCR revealed that mRNA expression of the autophagy-related genes, that is ATG3, ATG8b, ATG12 genes, were at the basal level throughout the treatment while ATG16 mRNA expression was increased at 18 and 24 h post-treatment (Fig. 3D).

## Effect of curcumin on *A. triangularis* encystation-related genes under a nutrient-rich condition

Apart from autophagy, we further assessed other *A. triangularis* encystation-related genes, namely, cellulose synthase (CS) and serine proteinase (SP) upon curcumin treatment. The mRNA expression pattern of both genes was similar in that expression was slightly changed during the first 18 h, and significantly increased at 24 h post treatment (Fig. 4).

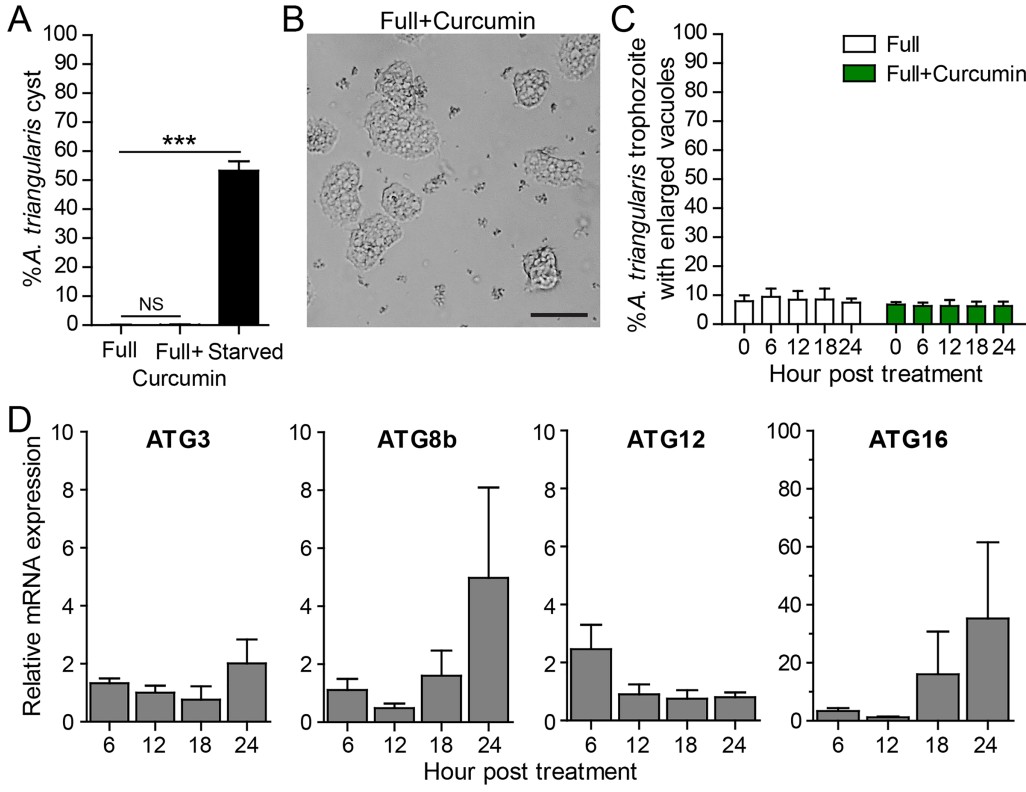

**Figure 3** *A. triangularis* **response to curcumin under a nutrient-rich condition.** (A) Cyst formation, the *Acanthamoeba* trophozoites were cultured in a PYG medium with or without 50 µg/mL curcumin for 24 h. Starvation was included as a positive control for cysts formation. Cysts were quantified every 6 h post treatment. A percentage of cysts was calculated and represented as mean ± SD. Data obtained from three independent experiments. NS, not significant; ***P < 0.001. (B) Representative image of curcumin-treated parasites under a full condition. Bar 20 µm. (C) Vacuolization in surviving trophozoites, at least 100 cells the trophozoites per condition were examined for enlarged vacuoles, a diameter of at least 5 µm. Data obtained from three independent experiments and represented as a mean percentage ± SD. NS, not significant. (D) Transcriptional expression of autophagy-related genes after curcumin treatment, *A. triangularis* trophozoites were cultured in PYG medium with or without 50 µg/mL curcumin for 24 h. The parasites were harvested every 6 h and the mRNA level of ATG3, ATG8b, ATG12, ATG16 genes were analyzed by qPCR. Their expression at each time point was expressed as a relative mRNA expression. 18S rRNA was included as a reference gene. The expression at time 0 h was set to 1. Data were obtained from three independent experiments. Bar graphs represent mean ± SEM.

Moreover, due to a crosstalk between autophagy and apoptosis in other eukaryotic cells, we also observed mRNA expression of genes involved in the apoptosis pathway, namely, metacaspase (MCA) and interleukin-1 converting enzyme-like protease (IL) in response to curcumin. MCA mRNA expression was slightly changed, but still at the basal level along with the treatment. The expression of IL mRNA was rapidly increased at 6 h post-treatment and gradually declined at later time points. However, the increase expression was again observed at 24 h post-treatment (Fig. S6). The response of surviving *A. triangularis* to curcumin either under nutrient starvation or nutrient-rich condition was illustrated in Fig. 5.

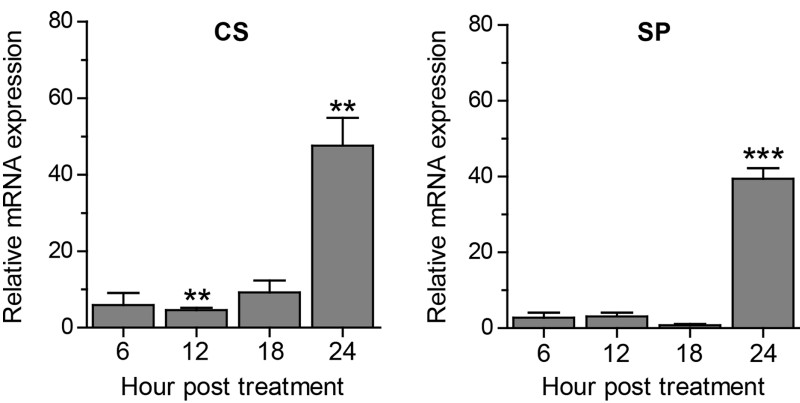

**Figure 4 Transcriptional expression of other encystation-related genes of *A. triangularis* in response to curcumin under a nutrient-rich condition.** Investigation of cellulose synthase (CS) and serine proteinase (SP) mRNA expression was carried out. cDNA samples were shared with autophagy analysis. The qPCR was performed, and 18S rRNA was included as a reference gene. Data were obtained from three independent experiments. Bar graphs displayed mean ± SEM. **$P < 0.01$; ***$P < 0.001$.

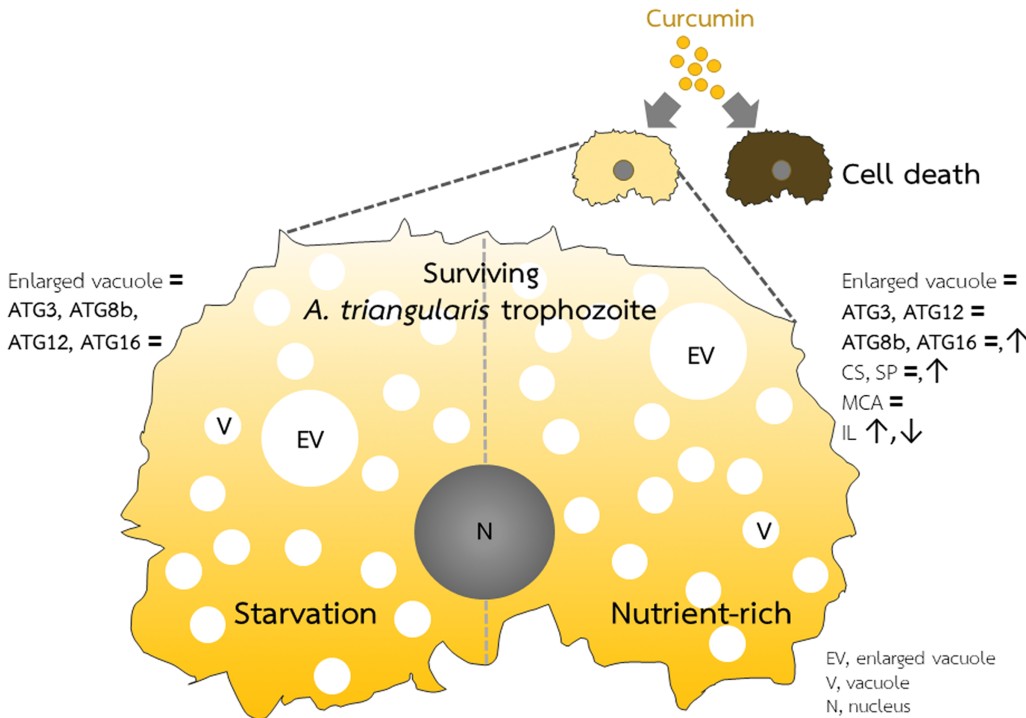

**Figure 5 *A. triangularis* response to curcumin.** Curcumin at the sublethal dose was used for *A. triangularis* trophozoites treatment. Approximately 50% of the parasites died, while the surviving parasites remained in the trophozoites stage. Transcriptional expression of tested autophagy-related genes was at the basal level in both starvation and a nutrient-rich condition except *Ac*ATG16, which increased at later time points under the nutrient-rich condition. Other *A. triangularis* encystation-related genes tested in this study, cellulose synthase and serine proteinase, were also increased at a later time point. Inducing an arrest in the trophozoites by curcumin is possibly resulting in the deactivation of the ATG genes and subsequent inhibition of vacuoles maturation.

### Curcumin-based drug combination study

A drug combination study between curcumin and chlorhexidine was performed, and our previous results on the co-treatment of curcumin and autophagy inhibitors, 3MA and wortmannin, did not completely inhibit *A. triangularis* encystation, thus these combinations were included in this assay to explore their interaction. The concentration of compound/drug was varied based on their MICs, except for the autophagy inhibitors which were designed to cover the concentration used in the previous experiment. The MICs of curcumin and chlorhexidine were started at 250 and 16 μg/mL, respectively. These were used as a starting concentration in the drug combination assay while the starting concentration of 3 MA and wortmannin was used at 20 mM and 20 μM, respectively. The results of parasite viability were represented as mean ± SD. In the curcumin-chlorhexidine combination assay, at maximum concentrations of curcumin (MIC 250 μg/mL) and chlorhexidine (MIC 16 μg/mL), the percentage of trophozoites viability was in the range of 5–8%. Reduction concentration of chlorhexidine to 8 μg/mL in combination with different concentrations of curcumin, the percentage was increased to the range of 42–53%, but their percentages were similar to those of chlorhexidine alone, at 52%. At lower concentrations of chlorhexidine (4, 2, 1 μg/mL), a pattern of the percentage viability at certain concentration of chlorhexidine was similar, and its percentage viability was gradually increased when the curcumin concentration was reduced (Table S1). In the curcumin-3MA (Table S2) and curcumin-wortmannin (Table S3) combinations, the pattern of results was similar. At certain 3MA or wortmannin concentrations below the MIC of curcumin, the percentage viability was gradually increased by reducing the concentration of curcumin. For the combinations that were close to our interest, 62.5 μg/mL curcumin-1.25 mM 3MA or 1.25 μM wortmannin, the percentage viability was comparable to curcumin alone.

## DISCUSSION

The cystic stage of *Acanthamoeba* is one of the main obstacles for therapeutic use as the penetration of anti-*Acanthamoeba* drugs across a double-layered cyst wall is fairly difficult (*Turner et al., 2000*; *Abjani et al., 2016*). The identification of new active compounds and drug repurposing with amoebicidal activity are urgently needed. In addition, the compound/drug that is able to prolong the trophozoite stage may be useful for drug combination purposes in the therapy of *Acanthamoeba* infection. In this study, an IC$_{50}$ of curcumin against *A. triangularis* was identified. The killing activity of curcumin was confirmed, and interestingly, the surviving amoebas were arrested in the trophozoite stage after curcumin treatment at the sublethal dose. The dual benefits of the curcumin, amoebicidal activity, and arresting cyst transformation against *Acanthamoeba* sp. gain more attention. Regarding a long history of medicinal use of curcumin, it contains several pharmacological activities, for example, anti-inflammatory (*Wal et al., 2019*), antioxidant (*Jakubczyk et al., 2020*), anti-cancer (*Vallianou et al., 2015*; *Tomeh, Hadianamrei & Zhao, 2019*), and antimicrobial activities (*Cui, Miao & Cui, 2007*; *Martins et al., 2009*; *Teow et al., 2016*; *Yang et al., 2016*; *Mitsuwan et al., 2020*). For its anti-parasitic effect, the curcumin has been very-well studied in many parasites for example

*Schistosomiasis mansoni* (*de Paula Aguiar et al., 2016*; *Hussein et al., 2017*), *Besnoitia besnoiti* (*Cervantes-Valencia et al., 2019*), *Giardia lamblia* (*Gutiérrez-Gutiérrez et al., 2017*), *Leishmania major* (*Koide et al., 2002*), *Plasmodium falciparum* (*Cui, Miao & Cui, 2007*; *Mishra et al., 2008*), and *Trypanosoma cruzi* (*Novaes et al., 2016*). However, the killing mechanism by curcumin has been partially characterized in some parasites, but in *Acanthamoeba*, the documentation is largely unidentified.

Encystation refers to a mechanism in which amoeba trophozoites are transformed into cysts under stress conditions (*Schaap & Schilde, 2018*). In *Acanthamoeba*, several pathways, for example, actin dynamics, glycolysis, proteolysis (*Bouyer et al., 2009*), proteins such as cyst specific protein 21 (*Chen et al., 2004*), serine protease (*Dudley, Alsam & Khan, 2008*; *Moon et al., 2008*), cysteine protease (*Leitsch et al., 2010*; *Moon et al., 2012*), glycogen phosphorylase (*Lorenzo-Morales et al., 2008*), sirtuin proteins (*Joo et al., 2020*), and Shwachman-Bodian-Diamond syndrome protein (*Wang, Lin & He, 2021*) have been reported to be involved with this mechanism. However, the coordination and crosstalk between these pathways to support the encystation are still unknown. Single or multiple pathways may be required for cyst formation and the cyst induction probably depends on the strength and specificity of the cyst formation signal. Autophagy is an intracellular stress-sensing mechanism that occurs rapidly in response to stimuli such as rapamycin, starvation, or cytokines (*Kamada, Sekito & Ohsumi, 2004*; *Kroemer, Mariño & Levine, 2010*). So far, more than 30 autophagy-related (Atg) proteins have been identified in yeast and humans, and their roles in this pathway have been extensively studied (*Kamada, Sekito & Ohsumi, 2004*; *Feng et al., 2014*; *Galluzzi et al., 2017*). However, a partial list of Atg proteins has been characterized in *Acanthamoeba i.e.* Atg3 (*Moon et al., 2011*), Atg8 (*Moon et al., 2009*; *Moon et al., 2013*), Atg12 (*Kim et al., 2015*), Atg16 (*Fujita et al., 2008*) and they were reported to be associated with *Acanthamoeba* encystation.

Starvation or a nutrient-depleted condition is a classical autophagy inducer in several eukaryotic cells (*Kamada, Sekito & Ohsumi, 2004*; *Mizushima et al., 2004*; *Mejlvang et al., 2018*). In *Acanthamoeba*, starvation conditions are able to induce *Acanthamoeba* encystation at different degrees, depending on the medium formulation, time, and *Acanthamoeba* spp. (*Aqeel et al., 2013*; *Sohn et al., 2017*; *Boonhok et al., 2021a*). In our study, starvation by Page's Saline buffer (PAS) supplemented with 5% glucose was utilized. Approximately 40–50% of cysts were observed at 24 h after initiation of the culture, and the percentage of trophozoites containing enlarged vacuoles was increased significantly. Autophagy inhibitors, 3MA, and wortmannin, which are known to inhibit phosphatidylinositol 3-kinase (PI3K) activity in the autophagy pathway (*Wu et al., 2010*), have also been applied to see an autophagic response in *A. triangularis*. The 3MA significantly inhibited the formation of *A. triangularis* cysts while wortmannin was slightly affected. The different degrees of inhibition may result from the specificity of binding to its PI3K substrate, and the concentration used in the assay. In the presence of curcumin at the sublethal dose under starved conditions, most of the parasites remained in the trophozoite, not transforming into cyst stages. However, the mechanism of action of curcumin remains to be elucidated and needs further investigation. The curcumin may

bind to *Acanthamoeba* surface or intracellular proteins including cell cycle proteins and autophagy-related proteins which leads to the cell cycle arrest and inhibition of encystation, respectively. Basically, induction of autophagy, a double membrane autophagosome or vacuole is formed (*Huang & Klionsky, 2002*; *Nakatogawa et al., 2009*), and in *Acanthamoeba*, the formation of vacuoles including autophagosome and autolysosome is associated with the cyst wall formation (*Bowers & Korn, 1969*). The percentage of trophozoites containing vacuoles or enlarged vacuoles was thus analyzed by microscopy in our study. Due to a highly active trophozoite stage (*Alves et al., 2017*), analysis of the trophozoites containing vacuoles, almost 100% of the trophozoites contained vacuoles, and the percentage of trophozoites containing vacuoles regardless of the vacuole size made a difference between tested conditions. However, analyzing trophozoites containing enlarged vacuoles, the percentage was significantly reduced upon autophagy inhibitor or curcumin treatment. However, the combination of curcumin with autophagy inhibitor did not completely inhibit cysts formation. This may result from the dose of drug/compound tested in this study. Moreover, our drug combination data also demonstrated no synergistic, additive, or antagonistic effects in any drug combinations against *A. triangularis* trophozoites. This indicates that the outcome observed in our study is derived from a single drug. Regarding the effect of curcumin under microscopic examination which markedly inhibited cysts formation and reduced vacuolization in surviving trophozoites, molecular analysis of *A. triangularis* autophagy mRNA expression was performed to assess a physiological change, in response to the curcumin. Considering *Acanthamoeba* autophagy, Moon and his colleague first characterized Atg8 in *Acanthamoeba castellanii* (*Moon et al., 2009*). *Ac*Atg8 was distributed in the amoeba cytosol, and its expression was peaked during encystation. In addition, intracellular colocalization of *Ac*Atg8 and lysosome on the membrane has been demonstrated (*Moon et al., 2009*). An *Ac*Atg8 isoform, *Ac*Atg8b was later identified. This isoform was highly expressed during encystation and was required for *Acanthamoeba* encystation (*Moon et al., 2013*). Atg3, an E2 ubiquitin-like conjugating enzyme, is known to play a role in the Atg8 conjugation system (*Feng et al., 2014*). In *A. castellanii*. *Ac*Atg3 was investigated by Moon and his colleagues and found that its mRNA expression was not increased during the encystation, but the depletion of *Ac*Atg3 affected the maturation of cysts (*Moon et al., 2011*). Atg12 plays a role in autophagosome formation by forming an Atg12-Atg5-Atg16L1 complex and acting as an E3-like enzyme to promote Atg8 lipidation in the autophagosomal membrane (*Yin, Pascual & Klionsky, 2016*). At the early phase of encystation, *Acanthamoeba* Atg12 was consistently distributed in trophozoites. Later, it was formed as a puncta and co-located with an autophagic membrane. Even its mRNA expression was not increased during encystation as expected, but it was crucial for the encystation as the down-regulation of *Ac*Atg12 in trophozoites inhibited cyst formation (*Kim et al., 2015*). *Acanthamoeba* Atg16 was partially colocalized with autophagolysosome and highly expressed during *A. castellanii* encystation (*Song et al., 2012*). Depletion of *Ac*Atg16 inhibited the formation of autophagosomes and further disrupted the encystation mechanism (*Song et al., 2012*). As expected, all tested genes were at the basal level in 3MA-or curcumin-treated conditions. The inhibition of key ATG mRNA expression thus

supports the attenuation of *A. triangularis* encystation as well as cyst production. Autophagy is a tightly regulated pathway and its response depends on the strength and specificity of signals (*Kroemer, Mariño & Levine, 2010*; *Simon et al., 2017*), thus our data indicate that the signal strength of curcumin is higher than starvation signal and its underlying mechanism, curcumin may specifically interact with proteins associated with the inhibition of *Acanthamoeba* encystment or cell cycle arrest (*Bínová, Bína & Nohtfytfnková, 2021*).

Next, we investigated the effect of a single curcumin signal under a nutrient-rich or full condition using PYG medium. The cyst formation in response to curcumin was at the basal level similar to full medium alone. The percentage of surviving trophozoites with enlarged vacuoles was also at the basal level and was not different between curcumin-treated and untreated conditions. In addition, the real-time PCR analysis revealed that the tested ATG genes were similar to those of curcumin treatment under starved condition except *Ac*ATG16 that up-regulated at later time points. The increased expression of *Ac*ATG16 was also observed in *Peganum harmala* seed extract-treated *A. triangularis* (*Boonhok et al., 2021a*); however, in *Cassia angustifolia* extract treatment, the increase in *Ac*ATG16 mRNA was not observed (*Boonhok et al., 2021b*). This may indicate a role of Atg16 in *A. triangularis* responses to the specific stress signal in autophagy or other cellular pathways, which requires further investigations. Under the nutrient-rich condition, the mRNA expression of other *A. triangularis* encystation-related genes was investigated. Both cellulose synthase (EDCBI66TR) and serine proteinase (EU365404) were slightly changed in the first 18 h, to our surprise, at 24 h, their expression was significantly increased even the microscopic examination showed that there was no cyst induction at this time. Cellulose is the main component of cyst wall, and three enzymes namely, glycogen phosphorylase, UDP-glucose pyrophosphorylase, and cellulose synthase, are required for cellulose synthesis during *Acanthamoeba* encystation (*Moon & Kong, 2012*; *Garajová et al., 2019*). In addition to cellulose synthase, investigation on the mRNA expression of another two genes is required to predict cyst formation after 24 h conclusively; otherwise, this may indicate an additional function of the cellulose synthase. On the other hand, serine proteinase that increased at 24 h post curcumin treatment may indicate its role in other cellular activities in addition to cell differentiation (*Blaschitz et al., 2006*; *Rascon & McKerrow, 2013*). Moreover, we observed metacaspase, which is known to be involved in apoptosis-like cell death in several microorganisms and associated with *A. castellanii* encystation (*Trzyna, Legras & Cordingley, 2008*; *Saheb, Trzyna & Bush, 2014*), as well as interleukin-1 converting enzyme-like protease, known as caspase-1, has a role in programmed cell death of parasites (*Kosec et al., 2006*; *Wu et al., 2018*). The mRNA expression of metacaspase (AF480890) was consistent over the time period of curcumin treatment which may support no cyst formation. However, interleukin-1 converting enzyme-like protease (XM004338552) was a quick response to curcumin as its mRNA expression was immediately increased at 6 h post-treatment. However, at later time points, its expression was declined to the basal level. The increase of this gene at an early time point may indicate an apoptotic cell death by curcumin. However, to confirm this type of cell death, an apoptosis assay is required. Once the amoeba is able to cope with

the curcumin stress, the interleukin-1 converting enzyme-like protease expression is gradually declined, which reveals an ability of *A. triangularis* trophozoites to overcome the curcumin stress or a death signal.

In the line of curcumin effect on autophagy, curcumin is known to modulate autophagy (*Shakeri et al., 2019*), and the outcome is varied depending on cell type and curcumin concentration as described herein. In human endothelial cells, EA.hy926 and HUVECs, 5 or 20 µM curcumin induced autophagy to reduce oxidative stress-induced cell damage (*Han et al., 2012*; *Guo et al., 2016*). An amount of 40 µM of curcumin was able to induce autophagy which is partially involved with anticancer activity in human lung adenocarcinoma cell line, A549 (*Liu et al., 2017*). In human colon cancer cells, HCT116, 40 µM curcumin-induced reactive oxygen species (ROS) production, which further activated autophagy followed by cell death (*Liu et al., 2017*). On the other hand, in mouse hippocampal neuronal cell line, HT-22, 10 or 15 µM curcumin promoted cell recovery in Aβ1-42-treated condition by inhibiting autophagy (*Zhang et al., 2018*). At 5 µM curcumin, it reduced apoptosis and inhibited autophagy and hypoxia-inducible factor 1-alpha in rat adrenal pheochromocytoma cell, PC12, model of oxygen-glucose deprivation/reperfusion (OGD/R) condition (*Hou, Wang & Feng, 2019*). Along with the OGD/R model, 10 µM curcumin was able to increase the resistance of cortical neurons by reducing autophagy and cell apoptosis in an mTOR-dependent manner (*Shi et al., 2019*). Even autophagy is a quick response to various stimuli, but its mechanism is tightly regulated and be more selective in which Atg proteins work together in a specific manner and coordinate with other pathways or proteins to create a wide variety of physiological processes in cells (*Wang & Qin, 2013*; *Galluzzi et al., 2017*) and the autophagic response might be varied depending on the cell type and the dose of curcumin. Investigation of function and physiological change of *A. triangularis* Atg proteins in response to stresses, including the curcumin stress, is needed. Regarding the ability of curcumin in cell arrest, several studies have mentioned this pharmacological activity. Curcumin treatment caused cell cycle arrest at G1/S and G2/M phases and activated a caspase-3 pathway, resulting in human osteosarcoma (HOS) cell death (*Lee, Lee & Kim, 2009*). In human cervical carcinoma cells, SiHa cells, curcumin activated ROS production, apoptosis, autophagy, cell cycle arrest, and cellular senescence. These activities co-occurred with the upregulation of p53 and p21 proteins (*Wang & Wu, 2020*). In colon cancer cells, HT-29, curcumin-induced ROS production led to apoptotic cell death and cell cycle inhibition (*Agarwal et al., 2018*). The similar results were observed in another colon cancer cell line, MC38. The mechanism of action of curcumin was also partially characterized and shown to down-regulate several cell cycle proteins *i.e.* cyclin A2, cyclin E1, cell cycle dependent kinase 2 (CDK2), and transcription factor E2F1 (*Li et al., 2022*). Altogether, *Acanthamoeba* autophagy and/or cell cycle pathway may be involved in our finding.

Curcumin and curcumin derivatives have so far been extensively studied for therapeutic purposes, especially in parasitic infections (*Din et al., 2016*). A successful development of a new class of curcumin has been reported against *Trypanosoma cruzi* (*Matiadis et al., 2021*). In *Plasmodium* infection, several strategies have been developed to increase the effectiveness of curcumin for example nanotized curcumin (*Ghosh et al., 2014*), curcumin

containing liposomes (*Martí Coma-Cros et al., 2018*), among others. The strategies open another direction in drug development that could be applied in *Acanthamoeba* research. Moreover, drug combination strategy by targeting the autophagy pathway in other models has been reported (*Zanotto-Filho et al., 2015*), and this strategy may applied in *Acanthamoeba* infection in the future. Taken together, evaluation of *Acanthamoeba* cyst formation and analysis of expression of autophagy-related genes or proteins in the surviving amoebas during drug or natural compound screening may help to assess the risk of *Acanthamoeba* encystation and can be a useful information for drug combination study to improve therapeutic efficacy and help reducing the drug resistance cycle in the area of infectious diseases (*Hill & Cowen, 2015*).

## CONCLUSIONS

Curcumin has a wide range of pharmacological activities and medicinal properties against numerous diseases. In *A. triangularis*, an amoebicidal activity of curcumin was recently demonstrated. Our study revealed that curcumin at sublethal dose is able to inhibit a transformation of *A. triangularis* trophozoites into cysts even under nutrient starvation. This may result from an attenuation of *Acanthamoeba* autophagy. However, an underlying mechanism of curcumin in *A. triangularis* trophozoites arrest is still unknown and needs further investigation. Overall, a dual benefit of curcumin, amoebicidal activity and arresting cyst transformation, may be another evidence to support drug development and future use of curcumin in *Acanthamoeba* infection therapy.

## ACKNOWLEDGEMENTS

We thank the Research Institute of Health Science (RIHS) staff at Walailak University.

### Funding

This research was funded by Walailak University grant No. WU-IRG-63-073, The Royal Patronage of Her Royal Highness Princess Maha Chakri Sirindhorn-Botanical Garden of Walailak Univer-sity, Nakhon Si Thammarat, under the project entitled: Medicinal Thai Native Plants against Acanthamoeba triangularis as a serious eye infection (WUBG 031-2565), Thailand. M.d.L.P. thanks to Project CICECO-Aveiro Institute of Materials, UIDB/50011/2020, UIDP/50011/2020, and LA/P0006/2020, financed by national funds through the FCT/MEC (PIDDAC). The funders had no role in study design, data collection and analysis, decision to publish, or preparation of the manuscript.

### Grant Disclosures

The following grant information was disclosed by the authors:
Walailak University: WU-IRG-63-073 and WUBG 031-2565.
Project CICECO-Aveiro Institute of Materials: UIDB/50011/2020, UIDP/50011/2020, and LA/P0006/2020, FCT/MEC (PIDDAC).

## Competing Interests

The authors declare that they have no competing interests.

## Author Contributions

- Rachasak Boonhok conceived and designed the experiments, performed the experiments, analyzed the data, prepared figures and/or tables, authored or reviewed drafts of the article, financial support, and approved the final draft.
- Suthinee Sangkanu conceived and designed the experiments, performed the experiments, analyzed the data, prepared figures and/or tables, and approved the final draft.
- Suganya Phumjan performed the experiments, analyzed the data, prepared figures and/or tables, and approved the final draft.
- Ramita Jongboonjua performed the experiments, analyzed the data, prepared figures and/or tables, and approved the final draft.
- Nawarat Sangnopparat performed the experiments, analyzed the data, prepared figures and/or tables, and approved the final draft.
- Pattamaporn Kwankaew performed the experiments, analyzed the data, prepared figures and/or tables, and approved the final draft.
- Aman Tedasen performed the experiments, analyzed the data, prepared figures and/or tables, and approved the final draft.
- Chooi Ling Lim conceived and designed the experiments, authored or reviewed drafts of the article, and approved the final draft.
- Maria de Lourdes Pereira conceived and designed the experiments, authored or reviewed drafts of the article, financial support, and approved the final draft.
- Mohammed Rahmatullah conceived and designed the experiments, authored or reviewed drafts of the article, and approved the final draft.
- Polrat Wilairatana conceived and designed the experiments, authored or reviewed drafts of the article, financial support, and approved the final draft.
- Christophe Wiart conceived and designed the experiments, authored or reviewed drafts of the article, and approved the final draft.
- Karma G. Dolma conceived and designed the experiments, authored or reviewed drafts of the article, and approved the final draft.
- Alok K. Paul conceived and designed the experiments, analyzed the data, prepared figures and/or tables, authored or reviewed drafts of the article, and approved the final draft.
- Madhu Gupta conceived and designed the experiments, authored or reviewed drafts of the article, and approved the final draft.
- Veeranoot Nissapatorn conceived and designed the experiments, authored or reviewed drafts of the article, financial support, and approved the final draft.

## Data Availability

The raw data are available in the Supplemental File.

## Supplemental Information

Supplemental information for this article can be found online at http://dx.doi.org/10.7717/peerj.13657#supplemental-information.

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
