# Peer review of "Curcumin effect on Acanthamoeba triangularis encystation under nutrient starvation"

_PeerJ, doi:10.7717/peerj.13657_

## Round 0.1 · original submission · Major Revisions

The review process is now complete, and two thorough reviews from highly qualified referees are included at the bottom of this letter. Although there is merit in your paper, we identified several concerns that must be considered in your resubmission.

1-Please, consider reviewing the abstract aiming to emphasize, in a direct manner, the importance and novelty of the study. Restructure the abstract topics according to their relevance and aims, and avoid misplaced information such as conclusions in MM and Results. Include information related to the Material and Methods used and a pertinent Conclusion. In the same way, the sections of the manuscript must be reorganized to avoid unnecessary repetitions.

2-The English language needs to be improved to make the manuscript clear for readers.

Reviewer 1 ·

Basic reporting

Article is well written. Sufficient literature is provided.

Experimental design

Study is well designed. Please provide justification for use of 100% DMSO as solvent for curcumin preparation. DMSO can interfere with study results, as reported literature indicate:
- DMSO is reported to induce apoptosis at concentrations >10% (v/v)
- DMSO stimulate prompt Acanthamoeba differentiation into a rounded cyst-like stage with a single envelope
- DMSO increases penetration of drugs into parasite cysts
- DMSO is cytotoxic

Validity of the findings

Findings are reported nicely. Please provide suitable justification to clarify that results obtained in this study are because of curcumin use only and there is no interference in study results with the use of DMSO and if DMSO interferes with study data, then please include role of DMSO also in discussion and conclusion.

Reviewer 2 ·

Basic reporting

The manuscript describes the effect of sub-lethal doses of curcumin on Acanthamoeba encystation, by investigating morphological parameters and expression of genes related to encystation. In its essence, the study proposal is sound because encystation inhibition is considered an auxiliary strategy for treating infections by Acanthamoeba. The article is well structured scientifically with adequate figures, the author presented consistent supplementary material and raw data. A robust set of references are presented, but some of them are dispensable. English writing makes understanding difficult in some sections. The text is wordy and an overall revision of the English language must be done. I strongly recommend the assistance of a colleague who is proficient in English or from a professional editing service.

Experimental design

The experimental approach involves the comparison of trophozoites treated or not with sublethal doses of curcumin, in starvation or nutrient-rich conditions, regarding morphological parameters and expression of encystation-related genes. The idea is original and based on previous evidence of the curcumin effect on Acanthamoeba inactivation. An English language revision would improve the overall quality of the text to make clear the main question. Some specific issues in methodology with impact even in the title should be addressed (see specific comments)

Validity of the findings

The data supported the conclusion that curcumin inhibits Acanthamoeba encystation, but the link with Atg genes was not confirmed. The conclusion description should be more assertive and concise in this direction.

Additional comments

Specific comments and questions are the following:
Main issues:
1. Abstract: In the Background subsection, the term “surviving amoebas” does not make sense if one does not know they were treated. Consider including “amoebas treated with sublethal doses of curcumin”. The aim of the study is not clear and assertive, I suggest rewriting mainly the last two phrases in the Background subsection.
2. Introduction: What do the authors consider the focus of the work? The inhibition of encystation by curcumin or the Atg genes expression? As written, the introduction section makes the central aim of the work ambiguous and does not properly connect these two topics. To my understanding, the curcumin effect on encystation is the central point, and the investigation of Atg and other genes expression will give insights into the encystation mechanisms. Therefore, the second paragraph's idea should be before the third one, and the connection of both should be improved.
3. Material and Methods: The encystation inductor medium (PAS+5% glucose) was based on Aqeel et al (2003) as the authors said. However, Aqeel et al used PBS + 10% glucose + 50mM MgCl2. If a modification was inserted, it must be described, as well as the reason for the modification.
4. Material & Methods: Does the term “starvation” adequate, considering that glucose is a nutrient able to support trophozoites activity for the period used in the assays (24h)? Besides, it is clear in the work of Aqeel et al (2003) that high osmolarity is the trigger for encystation. Thus, it seems conceptually incorrect to indicate starvation as the only inducer of encystation. Several recent and old works described encystation saline (no nutrients) to induce cyst formation by starvation as the Neff's encystation medium or other. (Examples: Rolland et al 2020, doi: 10.3390/pathogens9050321; Coulon et al. 2010, doi:10.1128/JCM.00309-10; Bowers & Korn 1969, doi:10.1083/jcb.41.3.786). Why don´t you use them instead of a glucose medium? This conceptual issue demands correction in the paper and in the title (I wonder if distinct types of encystation stimuli could influence the effect of curcumin and even the Atg genes expression, an approach for future investigations.)
5. Material & Methods: What do you mean by "irregular trophozoites"? Are they pleomorphic, presenting pseudopodia, acanthopodia, and vacuoles as active and healthy forms? If so, the term “irregular” can be misinterpreted.
6. Discussion: This section should be shortened to address some issues more objectively. Examples are descriptions on lines 383 to 392 and 516 to 541. Removing references and text restructuring will provide greater conciseness.
7. Conclusions: Pieces of information previously described in Introduction and Results are unproperly repeated here. I recommend rewriting to indicate the main findings according to the aim and the significance. It must be concise.
Additional minor comments:
8. Abstract: The methods should be described more technically.
9. Abstract: In the Results subsection, the description of the results should follow the same sequence as the main text.
10. Abstract: In the Results subsection, line 83: “Altogether, the data reveals that curcumin stress does not induce cysts formation…” Consider replacing by “curcumin stress inhibits cyst formation”
11. Introduction: A revision of the English language is needed. Several phrases can be improved in the style (Example on lines 78-85). What is “cidal”?? (line 129).
12. Material and Methods: The subsection “A. triangularis cultivation” should be the first.
13. Results: Tables S1, S2 and S3 do not indicate the evaluated parameter (percentage viability)
14. Results: Asterisks indicative of p-value significance are lacking in Atg 16 graphic. Include that transcriptional expression of other encystation-related genes is in response to curcumin in the caption of Figure 4.

---

## Round 0.2 · accepted · Accept

The authors have addressed most of the concerns raised by the reviewers.

Reviewer 1 ·

Basic reporting

No comment

Experimental design

No comment

Validity of the findings

No comment

---

## Author Rebuttal · Round 0.2

20<sup>th</sup> May 2022

Dear Editor,

We would like to thank Editor and reviewers' valuable comments to further improve the manuscript. Below is our point by point reply for your further consideration.

**Responses to Reviewer's Comments**
**Manuscript ID: 72176**
**Curcumin effect on *Acanthamoeba triangularis* encystation under nutrient starvation**

## Editor comments (Erika Braga)
**MAJOR REVISIONS**
The review process is now complete, and two thorough reviews from highly qualified referees are included at the bottom of this letter. Although there is merit in your paper, we identified several concerns that must be considered in your resubmission.

1-Please, consider reviewing the abstract aiming to emphasize, in a direct manner, the importance and novelty of the study. Restructure the abstract topics according to their relevance and aims, and avoid misplaced information such as conclusions in MM and Results. Include information related to the Material and Methods used and a pertinent Conclusion. In the same way, the sections of the manuscript must be reorganized to avoid unnecessary repetitions.
**Answers:** Thank you. The abstract, Materials and Methods, Results, Discussion, and Conclusion were revised.

2-The English language needs to be improved to make the manuscript clear for readers.
**Answers:** Thank you. The manuscript was proofread by a native speaker.

# Reviewer 1 (Anonymous)

*Basic reporting*

Article is well written. Sufficient literature is provided.

*Experimental design*

Study is well designed. Please provide justification for use of 100% DMSO as solvent for curcumin preparation. DMSO can interfere with study results, as reported literature indicate:
- DMSO is reported to induce apoptosis at concentrations >10% (v/v)
- DMSO stimulate prompt *Acanthamoeba* differentiation into a rounded cyst-like stage with a single envelope
- DMSO increases penetration of drugs into parasite cysts
- DMSO is cytotoxic

*Validity of the findings*

Findings are reported nicely. Please provide suitable justification to clarify that results obtained in this study are because of curcumin use only and there is no interference in study results with the use of DMSO and if DMSO interferes with study data, then please include role of DMSO also in discussion and conclusion.

**Answers:** Thank you for your positive comments. We apologize for a short explanation in the manuscript. We do agree that high %DMSO is toxic to the cell. In our study, we used 100%DMSO to prepare a stock curcumin. However, the working concentration of curcumin for *A. triangularis* trophozoites treatment is 50 µg/mL and the final %DMSO was 0.133%. However, during the identification of $IC_{50}$ of curcumin, the maximum concentration was 8,000 µg/mL and the final %DMSO was 2.13%. Therefore, the final concentration of DMSO used for cell treatment was not toxic and widely used in similar experiments elsewhere (clean version, lines 157-162).

# Reviewer 2 (Anonymous)

*Basic reporting*

The manuscript describes the effect of sub-lethal doses of curcumin on Acanthamoeba encystation, by investigating morphological parameters and expression of genes related to encystation. In its essence, the study proposal is sound because encystation inhibition is considered an auxiliary strategy for treating infections by *Acanthamoeba*. The article is well structured scientifically with adequate figures, the author presented consistent supplementary material and raw data. A robust set of references are presented, but some of them are dispensable. English writing makes understanding difficult in some sections. The text is wordy and an overall revision of the English language must be done. I strongly recommend the assistance of a colleague who is proficient in English or from a professional editing service.

**Answers:** Thank you. The reference section was revised. The manuscript was proofread by a native speaker.

*Experimental design*

The experimental approach involves the comparison of trophozoites treated or not with sublethal doses of curcumin, in starvation or nutrient-rich conditions, regarding morphological parameters and expression of encystation-related genes. The idea is original and based on previous evidence of the curcumin effect on *Acanthamoeba* inactivation. An English language revision would improve the overall quality of the text to make clear the main question. Some specific issues in methodology with impact even in the title should be addressed (see specific comments)

**Answers:** Thank you. The manuscript was revised.

*Validity of the findings*

The data supported the conclusion that curcumin inhibits *Acanthamoeba* encystation, but the link with Atg genes was not confirmed. The conclusion description should be more assertive and concise in this direction.

**Answers:** Thank you. The evidence to support the association between the inhibition of *A. triangularis* encystation and its autophagy needs further investigation. We are planning to work with *A. castellanii* in which its genome is available in NCBI to do 1) GFP tagging for localization study of Atg proteins in response to curcumin, 2) Molecular docking simulation to evaluate whether curcumin is able to bind to *Acanthamoeba* autophagy proteins including *Acanthamoeba* proteins involved in the cell cycle. This approach will provide some information on the inhibition of *Acanthamoeba* encystation and *Acanthamoeba* autophagy in response to curcumin.

*Additional comments*

Specific comments and questions are the following:

*Main issues:*

1. Abstract: In the Background subsection, the term "surviving amoebas" does not make sense if one does not know they were treated. Consider including "amoebas treated with sublethal doses of curcumin". The aim of the study is not clear and assertive, I suggest rewriting mainly the last two phrases in the Background subsection.

**Answers:** Thank you. The abstract was revised accordingly (clean version, lines 45-74).

2. Introduction: What do the authors consider the focus of the work? The inhibition of encystation by curcumin or the Atg genes expression? As written, the introduction section makes the central

aim of the work ambiguous and does not properly connect these two topics. To my understanding, the curcumin effect on encystation is the central point, and the investigation of Atg and other genes expression will give insights into the encystation mechanisms. Therefore, the second paragraph's idea should be before the third one, and the connection of both should be improved.

**Answers:** Thank you.  The introduction part was revised accordingly (clean version, lines 80-103, 124-139).

3. Material and Methods: The encystation inductor medium (PAS+5% glucose) was based on Aqeel et al (2003) as the authors said. However, Aqeel et al used PBS + 10% glucose + 50mM MgCl2. If a modification was inserted, it must be described, as well as the reason for the modification.

**Answers:** Thank you. The method section was revised (clean version, lines 149-151). The idea of using PAS+5%glucose was modified from Aqeel et al (2013). Regarding the medium used for *A. triangularis* cysts induction, we used Page's Saline (PAS) which contains similar constituents as the medium used by Aqeel et al 2013. However, in our lab, we work with natural isolated *A. triangularis* which is different from Aqeel et al 2003 that tested on *A. castellanii*. In the beginning, we prepared PAS with different percentages of glucose including without glucose conditions. We found that in PAS alone without glucose, some dead amoebas were observed at 24 h culture. In PAS+10%glucose, PAS+8%glucose, PAS+5%glucose, all amoebas were shown no different in term of still alive (viability) and the percentage of cysts concentration. Thus, PAS+5%glucose was chosen to be used in our study for cysts induction.

4. Material & Methods: Does the term "starvation" adequate, considering that glucose is a nutrient able to support trophozoites activity for the period used in the assays (24h)? Besides, it is clear in the work of Aqeel et al (2003) that high osmolarity is the trigger for encystation. Thus, it seems conceptually incorrect to indicate starvation as the only inducer of encystation. Several recent and old works described encystation saline (no nutrients) to induce cyst formation by starvation as the Neff's encystation medium or other. (Examples: Rolland et al 2020, doi: 10.3390/pathogens9050321; Coulon et al. 2010, doi:10.1128/JCM.00309-10; Bowers & Korn 1969, doi:10.1083/jcb.41.3.786). Why don´t you use them instead of a glucose medium? This conceptual issue demands correction in the paper and in the title (I wonder if distinct types of encystation stimuli could influence the effect of curcumin and even the Atg genes expression, an approach for future investigations.)

**Answers:** Thank you. In our work, the term starvation, we actually refer to amino acid starvation which is a main component that is different between PAS and PYG media.
Regarding the medium used for *A. triangularis* cysts induction, we used Page's Saline (PAS) which contains similar constituent as the medium used by Aqeel et al 2013. However, in our lab, we work with natural isolated *A. triangularis* which is different from Aqeel et al 2013 that tested on *Acanthamoeba castellanii*. In the beginning, we prepared PAS with different percentages of glucose including without glucose conditions. We found that in PAS alone without glucose, some dead amoeba was observed at 24 h culture. In PAS+10%glucose, PAS+8%glucose, all amoebas were shown no different in term of still alive (viability) and the percentage of cysts concentration (please refer to the Table below, bar 20µm). Thus, PAS+5%glucose was chosen to be used in our study for cysts induction.
We agree with the reviewer that different encystation stimuli may give a different cell response in particular in the presence of curcumin or any other compounds. However, in this study, we used

one type of starvation medium to induce *A. triangularis* encystation at 24 h, thus, to our results, we tried to summarize in the context of the medium, *A. triangularis*, and the time of treatment. Investigation of starvation media of different formulations for the encystation and their effect on autophagy of different *Acanthamoeba* species are of interest and will be a next step of our research plan.

| Medium | %Cyst (mean ± SD) | Remark | Representative image |
|---|---|---|---|
| PAS+ 10%glucose | 58.23 ± 3.38 | Boonhok, R, et.al. Pathogens 2021, 10, 842. https://doi.org/10.3390/pathogens10070842 |  |
| PAS+ 5%glucose | 53.30 ± 3.24 | Current study |  |

*A.triangularis at 24 h culture

5. Material & Methods: What do you mean by "irregular trophozoites"? Are they pleomorphic, presenting pseudopodia, acanthopodia, and vacuoles as active and healthy forms? If so, the term "irregular" can be misinterpreted.

**Answers:** Thank you. The trophozoites are pleomorphic and in our study, we would like to simply provide more information on the shape of trophozoites i.e. round shape and irregular shape. The term "irregular trophozoites" was obtained from https://doi.org/10.1111/jeu.12147 and https://doi.org/10.1016/j.aquaculture.2019.04.036 and in this study, we intend to refer to the healthy irregular-shaped trophozoites containing acanthopodia.

6. Discussion: This section should be shortened to address some issues more objectively. Examples are descriptions on lines 383 to 392 and 516 to 541. Removing references and text restructuring will provide greater conciseness.

**Answers:** Thank you. The discussion section was revised (clean version, lines 381-385, 393-395, 404-406, 410-414, 427-432, 442-445, 469-474, 526-532, 539-543, 548-558).

7. Conclusions: Pieces of information previously described in Introduction and Results are unproperly repeated here. I recommend rewriting to indicate the main findings according to the aim and the significance. It must be concise.

**Answers:** Thank you. The conclusion was revised (clean version, lines 561-569).

*Additional minor comments:*
8. Abstract: The methods should be described more technically.

**Answers:** Thank you. The abstract was revised.

9. Abstract: In the Results subsection, the description of the results should follow the same sequence as the main text.

**Answers:** Thank you. The abstract was revised.

10. Abstract: In the Results subsection, line 83: "Altogether, the data reveals that curcumin stress does not induce cysts formation…" Consider replacing by "curcumin stress inhibits cyst formation"

**Answers:** Thank you. The abstract was revised accordingly.

11. Introduction: A revision of the English language is needed. Several phrases can be improved in the style (Example on lines 78-85). What is "cidal"?? (line 129).

**Answers:** Thank you. The introduction section was revised accordingly. The cidal activity was replaced with killing activity.

12. Material and Methods: The subsection "*A. triangularis* cultivation" should be the first.

**Answers:** Thank you. We edited accordingly (clean version, lines 144-153).

13. Results: Tables S1, S2 and S3 do not indicate the evaluated parameter (percentage viability)

**Answers:** Thank you. We edited accordingly.

14. Results: Asterisks indicative of p-value significance are lacking in Atg 16 graphic. Include that transcriptional expression of other encystation-related genes is in response to curcumin in the caption of Figure 4.

**Answers:** Thank you.
In Figure 3, the analysis of mRNA expression was obtained from n=3. 2 out of 3 clearly demonstrated an increase of ATG16 mRNA expression. However, 1 out of 3, the fold change was close to 1. Thus, after statistical analysis of ATG16 mRNA expression, data showed no significance at both 18 and 24h post treatment.
The caption of Figure 4 was edited accordingly.

# Additional suggestions by PeerJ

1.Re-used Text
Lines 213-216 https://www.mdpi.com/2076-0817/10/7/842
Lines 217-222 https://www.cambridge.org/core/journals/parasitology/article/abs/amoebicidal-activity-of-cassia-angustifolia-extract-and-its-effect-on-acanthamoeba-triangularis-autophagyrelated-gene-expression-at-the-transcriptional-level/B9A3D96F05D946D5B94D305BCE00C3A9 (Methods, same author)
**Answers:** Thank you. The method was revised accordingly (clean version, lines 213-216).

2.Figure Style
For maximum figure accessibility, please remove the patterns/gradients in your bar graphs in Figures 2 and 3.
Please provide replacement figures measuring minimum 900 pixels and maximum 3000 pixels on all sides, saved as PNG, EPS or vector PDF file format without excess white space around the images.
**Answers:** Thank you. The figures were edited accordingly. All figures, Fig 1-5 and Fig S1-S6, were exported as PNG file with 300dpi.

3. Figure Quality
Please reformat Figures 2, 3, and 4 by removing the line break in the middle of your data range. Truncating the data range makes your measurements appear closer than they actually are.
Please provide replacement figures measuring minimum 900 pixels and maximum 3000 pixels on all sides, saved as PNG, EPS, or vector PDF file format without excess white space around the images here.
**Answers:** Thank you. All figures, Fig 1-5 and Fig S1-S6, were edited accordingly and exported as PNG file with 300dpi.

4.Abbreviations
Please define the first use of any abbreviation in the text. For example, "ATG" in the abstract (Line 63) needs to be spelled out at the first use.
**Answers:** Thank you. We edited accordingly.